# Comparing Visual and Software-Based Quantitative Assessment Scores of Lungs’ Parenchymal Involvement Quantification in COVID-19 Patients

**DOI:** 10.3390/diagnostics14100985

**Published:** 2024-05-08

**Authors:** Marco Nicolò, Altin Adraman, Camilla Risoli, Anna Menta, Francesco Renda, Michele Tadiello, Sara Palmieri, Marco Lechiara, Davide Colombi, Luigi Grazioli, Matteo Pio Natale, Matteo Scardino, Andrea Demeco, Ruben Foresti, Attilio Montanari, Luca Barbato, Mirko Santarelli, Chiara Martini

**Affiliations:** 1Department of Diagnostic Imaging, Spedali Civili di Brescia, Piazzale Spedali Civili 1, 25123 Brescia, Italy; 2Department of Neuroradiology, University Hospital of Padova, Via Giustiniani 2, 35128 Padova, Italy; 3Department of Radiological Function, “Guglielmo da Saliceto” Hospital, Via Taverna 49, 29121 Piacenza, Italy; 4Department of Radiology—Diagnostic Imaging, ASST Rhodense, Viale Forlanini 95, 20024 Garbagnate Milanese, Italy; 5Department of Respiratory Disease, University of Foggia, Via Antonio Gramsci 89, 71122 Foggia, Italy; matteo.natale@unifg.it; 6Department of Radiology, A.O.U. Città della Salute e della Scienza di Torino, Via Zuretti 29, 10126 Torino, Italy; mscardino11@gmail.com; 7Department of Medicine and Surgery, University of Parma, Via Gramsci 14, 43126 Parma, Italy; andrea.demeco@unipr.it (A.D.);; 8Diagnostics for Images Unit and Interventional Radiology, AST Pesaro Urbino, Piazzale Cinelli 1, 61121 San Salvatore, Italy; attilio.montanari@sanita.marche.it; 9Radiology Unit, Department of Medical Surgical Sciences and Translational Medicine, “Sapienza” University of Rome, Sant’Andrea University Hospital, Via Di Grottarossa, 1035-1039, 00189 Rome, Italy; 10Medical Physics Unit, “Sapienza” University of Rome, Sant’Andrea University Hospital, Via Di Grottarossa, 1035-1039, 00189 Rome, Italy; 11Diagnostic Department, Parma University Hospital, Azienda Ospedaliero-Universitaria di Parma, Via Gramsci 14, 43126 Parma, Italy

**Keywords:** chest CT, artificial intelligence, visual score, software-based score, COVID-19, deep learning

## Abstract

(1) Background: Computed tomography (CT) plays a paramount role in the characterization and follow-up of COVID-19. Several score systems have been implemented to properly assess the lung parenchyma involved in patients suffering from SARS-CoV-2 infection, such as the visual quantitative assessment score (VQAS) and software-based quantitative assessment score (SBQAS) to help in managing patients with SARS-CoV-2 infection. This study aims to investigate and compare the diagnostic accuracy of the VQAS and SBQAS with two different types of software based on artificial intelligence (AI) in patients affected by SARS-CoV-2. (2) Methods: This is a retrospective study; a total of 90 patients were enrolled with the following criteria: patients’ age more than 18 years old, positive test for COVID-19 and unenhanced chest CT scan obtained between March and June 2021. The VQAS was independently assessed, and the SBQAS was performed with two different artificial intelligence-driven software programs (Icolung and CT-COPD). The Intraclass Correlation Coefficient (ICC) statistical index and Bland–Altman Plot were employed. (3) Results: The agreement scores between radiologists (R1 and R2) for the VQAS of the lung parenchyma involved in the CT images were good (ICC = 0.871). The agreement score between the two software types for the SBQAS was moderate (ICC = 0.584). The accordance between Icolung and the median of the visual evaluations (Median R1–R2) was good (ICC = 0.885). The correspondence between CT-COPD and the median of the VQAS (Median R1–R2) was moderate (ICC = 0.622). (4) Conclusions: This study showed moderate and good agreement upon the VQAS and the SBQAS; enhancing this approach as a valuable tool to manage COVID-19 patients and the combination of AI tools with physician expertise can lead to the most accurate diagnosis and treatment plans for patients.

## 1. Introduction

Radiological imaging had a crucial part during the coronavirus disease 2019 (COVID-19) pandemic. Computed tomography (CT) played a paramount role in the characterization and follow-up of the illness, and its importance is broadly accepted [1,2].

In the early stages of the pandemic, when reliable and readily available tests like PCR tests were limited, CT scans provided valuable insights into lung involvement caused by COVID-19, and they were also used to monitor a patient’s response to treatment by tracking changes in lung abnormalities over time. 

Typical manifestations of COVID-19 pneumonia on chest CT images have been reported in various studies [3,4] such as ground-glass opacity (GGO), which is a non-specific term defined by the Fleischner Society as the presence on high-resolution computed tomography (HRCT) of a hazy increase in lung density, not associated with an obscuration of the underlying vessels or bronchial walls; when vessels are obscured, the proper term used is “consolidation” [5]. The hallmark of GGO is that the underlying blood vessels and bronchial walls remain visible, despite the increased lung density. Consolidation refers to a complete filling of the air spaces in the lungs with fluid or inflammatory cells. In this scenario, the underlying blood vessels and bronchial walls are obscured on CT scans due to the complete airspace filling. In addition, GGO generally suggests earlier stages of lung involvement, while consolidation might indicate more advanced inflammation or fluid accumulation. Crazy-paving pattern (CPP) is a term to use to describe a non-specific radiological sign that is characterized by the presence of diffuse ground-glass attenuation associated with interlobular septal thickening and intralobular lines. 

Various studies investigated the possibility of drafting a tailored low-dose chest CT protocol for infected patients, such as Homayounieh F et al. [6] who discussed this matter through a survey issued by the International Atomic Energy Agency (IAEA) from May to July 2020. The questionnaire collected data regarding scan parameters, dose-related information, having a dedicated CT protocol for COVID-19 patients, how many CT scanners were available in the facility and which type of CT protocol was the most used for this type of patient. The authors analyzed CT acquisition protocols across all the vendors. It resulted that a limit of CTDIvol (Volume CT Dose Index) less than 3 mGy (Gray) is acceptable when the evaluation is limited to lung parenchyma. Additionally, they encouraged to use iterative reconstruction properly to achieve a lower dose for infected patients. A systematic review conducted by Suliman, I.I. et al. [7] collected different low-dose chest CT protocols for COVID-19 through varied scientific databases. The authors gathered the scanning parameters from the main papers comparing the standard protocol (STD) versus the ultra-low-dose one (ULD). It has been enhanced as the following: lower kV, pitch higher than 1, using iterative reconstruction (IR), tube current modulation and fixed mAs were implemented to achieve the ULD protocol. 

During the different waves of the pandemic, the use of artificial intelligence (AI) emerged as a valuable tool to analyze chest CT scans and assess the illness severity. The main goals of this implementation were to quantify the lung parenchyma involvement more accurately, reduce the workload for radiologists and improve the efficiency of diagnosis and treatment decisions. Thus, several types of software were developed during the pandemic to help radiologists in the diagnosis of COVID-19, especially when lung CT was the most requested exam. 

This software has shown its utility to face an increased workload and to accelerate the process of diagnosis connected to this software [8]; several score systems had been implemented to properly assess the lung parenchyma involved in patients suffering from SARS-CoV-2 infection. They have been mainly divided into two methods: the visual quantitative assessment score (VQAS) and software-based quantitative assessment score (SBQAS). The first one relies on the amount of lung abnormalities visually recognized by experienced radiologists, while the second one is built upon software based on AI to automatically or semi-automatically detect lesions giving a report about the quantification of the lung parenchyma involved. 

Therefore, this study aims to compare the diagnostic accuracy between the visual quantitative and software-based assessment obtained from two different software types regarding the quantification of the lung parenchyma affected by SARS-CoV-2 infection to investigate the differences and the strong points within them, to establish their reliability.

## 2. Materials and Methods

### 2.1. Study Population

Approval for this study was granted by the local ethics committee (approval number NP5928). The institutional review board waived the requirement to obtain written informed consent for this retrospective case series, since all analyses were performed on de-identified data, therefore there was no potential risk to patients. 

A total of 90 patients were included with the following criteria: patients’ age more than 18 years old, real-time polymerase chain reaction (RT-PCR) test positive for COVID-19 and an unenhanced chest CT scan obtained between March and June 2021 at the Spedali Civili Hospital of Brescia, Italy. All patients that did not meet the following criteria were not included in the study: age, gender, weight, height, BMI (Body Mass Index) and clinical indication for chest CT were recorded at the time of the examination. 

### 2.2. CT Protocol

The entire population of this study underwent a chest CT scan without the injection of a contrast agent on a 64-detector scanner (Philips Brilliance 64; Amsterdam, The Netherlands). 

The scanning range was from the apex to the base of the lungs with the images obtained at full inspiration in the supine position. The chest CT parameters were as follows: kV range 100–140 kV, 80–350 effective mAs, using both z-axis and angular tube current modulation, fixed mAs 30–80 for a few patients (*n* = 11) (7), 0.4 s rotation time and pitch 0.8 to 1.2 (Table 1). All data were reconstructed using a sharp reconstruction kernel for parenchyma evaluation and the constructor’s iterative reconstruction iDose^4^ with a strength of 4 to 7. In the literature, a comparison between the application of different levels of strength of iDose4 showed a non-significant difference in the image quality and in their interpretation [9,10]. The window center and window width were set at −600 and −1600. There were no dedicated COVID-19 parameters for chest CT scans, resulting in using different strategies to achieve proper dose and image quality, such as fixed mAs and a higher level of IR (7).

Radiation doses were expressed in the Computed Tomography Dose Index (CTDI) and Dose-Length Product (DLP). The mean and median CTDI were, respectively, 8.23 ± 4.20 and 7.13, while the mean and median DLP were, respectively, 383.3 ± 208.88 and 342.

### 2.3. Visual Quantitative Assessment Score

The VQAS for each patient of this study was performed independently by two radiologists (S.P and M.L) with more than 10 years of experience.

CT images were independently reviewed and analyzed according to the Fleischner Society Glossary of terms for Thoracic Imaging [5]. The reviewers were also blinded from the clinical data, such as fever, cough, dyspnea and oxygen saturation, to reduce bias and ensure the internal validity of this study. Finally, the reviewers categorized CT findings as highlighted by Sverzellati et al. [11].

The VQAS was formulated according to some previous studies [12,13,14]. In particular, the two readers gave a percentage because of the lung parenchyma involved by COVID-19 following the criteria of the total severity score proposed by Li K. et al. [15]. This scoring system analyzes the affected parenchyma in each of the five lung lobes. Each lobe is assessed for the percentage of its volume affected by the disease and assigned a score based on the following scale: none (0% involvement) score 0, minimal (1–25% involvement) score 1, mild (26–50% involvement) score 2, moderate (51–75% involvement) score 3 and severe (76–100% involvement).

This approach was taken to minimize potential biases, such as inter-observer variability, and the efforts to minimize it were adapting a standardization of detailed scoring guidelines and a blinded scoring between the two readers. 

### 2.4. Software-Based Quantitative Assessment Score

The software-based assessment score (SBQAS) was performed with two different types of AI-based software. 

The first one, “Icolung” (Icometrix, Leuven, Belgium), is a cloud-based software that automatically contours the lungs. Moreover, it returns a report with the percentage of the lung parenchyma involved [16]. This software is based on deep learning (DL) models that sequentially carry out fully automated lung segmentation and lung abnormalities, such as ground-glass opacity (GGO), crazy-paving pattern (CPP) and consolidation. Deep learning algorithms are a type of machine learning inspired by the structure and function of the human brain. They achieve complex tasks by mimicking the way neurons connect and transmit information in artificial neural networks (ANNs). These are interconnected layers of processing units (artificial neurons) that loosely resemble biological neurons. 

Each layer receives input from the previous layer, performs a mathematical operation and sends its output to the next layer. DL excels at finding patterns in large amounts of data. During training, the algorithm is fed a massive dataset of labeled examples. Each example consists of an input and a corresponding output. Convolutional Neuronal Networks (CNNs) are a type of DL that is able to process image inputs, and they use convolutional layers to extract features from the input data. Its architecture reflects the connectivity pattern of neurons in the human brain. CNNs are trained on massive datasets of medical images that have been annotated with ground truth labels. These labels might indicate the presence or absence of specific abnormalities, the location of lesions or even specific disease types. The concept of “ground truth” refers to the confirmed and established diagnosis of a medical condition based on all available clinical information. In radiology, this typically involves the combined expertise of experienced radiologists, and it serves as benchmark for evaluating the accuracy of CNN predictions. High-quality ground truth annotations are essential for training effective CNNs in radiology.

The Icolung report shows the abnormalities visualized in the 2D axial and coronal view and a table with the total lung involvement percentages, divided for each lobe, and the corresponding severity scores (0–5 score per each lobe) based on Pan. F et al. [17]. The flowchart of Icolung’s framework is illustrated in Figure 1.

The second software used in this study is called Philips IntelliSpace Portal clinical application CT-COPD (Philips, Eindihoven, The Netherlands) computer tool. It is a semi-automatic software for lung segmentation; it was mainly used to measure the extent of the pulmonary emphysema in patients with chronic obstructive pulmonary disease (COPD). It enables us to set up a threshold of Hounsfield unit (HU) to quantify the lung parenchyma accordingly to the needs of the operator. In this study, the HU threshold chosen to establish the lung parenchyma affected by SARS-CoV-2 infection was set at -750, as proposed in other studies [8,18,19]. The SBQAS for this tool was performed by two blinded and trained radiographers (A.M, M.N), and the result was obtained by considering the percentage of the total lung parenchyma amount minus the extent of the percentage of aerated residual lung volume.

An example of the report obtained from Icolung and CT-COPD are, respectively, illustrated in Figure 2 and Figure 3.

### 2.5. Statistical Analysis

The statistical analysis was conducted using IBM SPSS (Statistical Package for the Social Sciences) version 29.0.1.0 (171) and Prism GraphPad version 9.5.1 to ensure comprehensive data analysis and accurate interpretation. Categorical variables were presented as counts and percentages, while continuous variables were expressed as medians. 

To assess agreement among the two radiologists regarding the VQAS for lung parenchyma involvement on the CT images, as well as between the software quantification, the Intraclass Correlation Coefficient (ICC) statistical index was employed. The ICC score ranges from −1 to 1, and for interpreting its values, the following criteria were utilized: values less than 0.50 were indicative of poor reliability, values ranging from 0.50 to 0.75 indicated moderate reliability, values ranging from 0.75 to 0.90 indicated good reliability and values greater than 0.90 indicated excellent reliability. This interpretation framework helped assess the level of agreement and reliability achieved in both the radiologists’ visual quantitative assessment score for disease extension and the software-based assessment score of the lung parenchyma involved.

This index was also used to analyze the SBQAS with the involvement of two distinct operators, whose assessments were blinded to the group allocations. This procedural refinement sought to mitigate the influence of individual operator biases. Notably, the ICC, a widely employed metric for assessing inter-rater reliability, was computed to quantify the degree of agreement between the independently derived scores.

By utilizing the ICC, it was possible to be able to quantitatively evaluate the degree of agreement and reliability among the raters or assessments, providing valuable insights into the consistency and concordance of their evaluations. 

## 3. Results

### 3.1. Patient Characteristics

A total of 79 patients were considered, with a mean age of 69 ± 12 years, ranging from a minimum of 37 to a maximum of 95. The interquartile range (IQR, 25° and 75°) was, respectively, 59 and 78 years. A total of 11 patients were excluded from the statistical analysis due to severe motion and respiratory artifacts, which could mimic lung abnormalities. This could lead the SBQAS to a miscalculation of lung parenchyma.

### 3.2. Inter-Reader Agreement of Visual and Software-Based CT Assessment

The agreement between radiologists (R1 and R2) for the visual quantitative assessment score of the lung parenchyma involved in the CT images was good (ICC = 0.871). The agreement between the two software programs (Icolung and CT-COPD) for the SBQAS was moderate (ICC = 0.584). The descriptive statistics and the boxplot of the two software programs and the radiologists are summarized, respectively, in Table 2 and Figure 4.

The agreement regarding the SBQAS showed a calculated ICC value of 0.95, which signifies a high level of concordance between the assessments provided by the two operators. This observation underscores the consistency and reliability of the SBQAS scores across different raters.

The agreement between Icolung and the median of the visual evaluations (median R1–R2) is good (ICC = 0.885). The agreement between CT-COPD and the median of the visual evaluations (Median R1–R2) is moderate (ICC = 0.622).

Interestingly, the second software, CT-COPD, has an overestimation of the results, as indicated by the higher median, first and third percentiles. Also, the first radiologist (R1) presents higher values (median, first and third percentiles) as compared to the second one (R2) (Figure 5).

In Figure 5, the first graph represents the assessment between the two software types Icolung and COPD; the results lie in a range between −47.73 and −2.78 with an SD of ±1.96. The second represents the comparison between the visual descriptions between radiologists (R1 and R2); the results lie in a range between −16.59 and 32.44 with an SD of ±1.96. The third graph shows the first software (Icolung) vs. the median of the visual estimations; the results lie in a range between −31.75 and 19.59 with an SD of ±1.96. The fourth graph shows the second software (COPD) vs. the median of the visual estimations; the results lie in a range between −18.41 and 56.75 with an SD of ±1.96.

## 4. Discussion

Managing COVID-19 patients by assessing the extent of the lung parenchyma involved was cardinal during the COVID-19 pandemic. Hence, AI resulted in a valid and helpful tool to assist physicians in this process as a decision-making aid and not a replacement for the expertise of medical professionals. 

This study has shown good agreement (ICC = 0.871) between the two blinded radiologists (R1 and R2) for the visual quantitative assessment score of the lung parenchyma involved in the CT images. This indicates a univocal method of lung parenchyma abnormality detection. Therefore, the ICC score suggests a consistent and univocal method for detecting lung parenchyma abnormalities. 

Additionally, the agreement between the two software types (Icolung and CT-COPD) for the SBQAS is moderate (ICC = 0.584). This result could be explained by analyzing the nature of these two different software types. Icolung is an automatic DL software trained during the pandemic, while CT-COPD is a software designed to quantify chronic obstructive pulmonary disease and adapted to evaluate the extension of the lung parenchyma affected by SARS-CoV-2. Moreover, the key point of this outcome might rely on the focused training of Icolung, which, being a DL software trained specifically during the COVID-19 pandemic, had likely been trained for a focused target such as identifying the lung abnormalities in patients affected by SARS-CoV-2. This process could lead to a more accurate report. Alongside this aspect, CT-COPD, originally designed for chronic obstructive pulmonary disease (COPD) as stated before, might have been adapted to assess COVID-19 lung involvement resulting in being not as fine-tuned as Icolung to recognize and detect specific patterns of SARS-CoV-2 infection. Finally, the difference in HU threshold selection might reflect the strength of DL for medical image analysis. 

In addition to this, it was found that the agreement between Icolung and the median of the visual evaluations (median R1-R2) is good (ICC = 0.885). This score suggests that Icolung’s measurements closely align with what experienced radiologists were seeing on the chest CT images. The agreement between CT-COPD and the median of the visual evaluations (median R1–R2) is moderate (ICC = 0.622), showing a less consistent agreement with human experts. This aspect outlines the validation of the AI-based software as a data-driven approach. 

Overall, it is worth mentioning that CT-COPD presents an overestimation of the results, as indicated by the higher median, first and third percentiles. This may rely on the possibility of editing the lung parenchyma contouring proposed by the software. 

These findings highlight a few points regarding AI-based software validation, such as the task-specific training matter, since Icolung’s strong agreement likely stems from being specifically trained to identify COVID-19 lung patterns. This targeted training allows it to perform well on the SBQAS task, and the DL advantage as the ability of its models to learn from vast amounts of datasets to identify subtle patterns in medical imaging that might be difficult for traditional programming methods to capture. 

The topic of this article has been investigated by several authors, each one of those with different peculiarities. Granata V. et al. [18] used the clinical application CT-COPD (Philips, Eindihoven, The Netherlands) to evaluate the critical lung involvement in patients vaccinated or unvaccinated affected by different variants of SARS-CoV-2, finding this tool suitable for pathological abnormalities, mainly regarding the assessment of consolidation. They calculated the disease severity by considering the percentage of aerated residual lung volume, and therefore, patients with lower aerated residual lung volume were considered more compromised. Good statistically significant correlations among volumes extracted by an automatic tool for each lung lobe and the overall radiological severity score were obtained (ICC range 0.71–0.86). Another study conducted by Durhan G. et al. [19] retrospectively assessed COVID-19 patients who underwent chest CT. The authors compared the VQAS with the normal lung parenchyma percentage made by a DL software, and they suggested that this parameter could give valuable and objective information about pneumonia due to the infiltrative nature of lung involvement. 

DL software implemented in radiology to evaluate patients affected by SARS-CoV-2 has been used since the COVID-19 outbreak. Saba L. et al. [20] compared six different AI paradigms, and the authors demonstrated that AI can automatically extract tissue features and characterize the disease, distinguishing between non-COVID-19 pneumonia and COVID-19 pneumonia. A clinical example of this model can be found in other studies, such as Suri J.S et al. [21] and Gujot J. et al. [22], in which the earlier cited software offered a valid tool to detect and classify affected patients. Nevertheless, Jungmann F. et al. [23] stated their concern regarding the actual AI solutions, such as Icolung, as tools to assess positive predictive value (PPV). The aim of this study was to evaluate the performance of commercial AI solutions in differentiating COVID-19 pneumonia from other lung conditions. The authors retrospectively collected 50 chest CT scans from two different tertiary care hospitals with proven SARS-CoV-2 infection. The visual assessment of the disease extent was performed according to CO-RADS (COVID-19 Reporting and Data System) [24], and the SBQAS was carried out by using four different companies, and Icolung was one of them. This article emphasized a low and variable specificity and low positive predictive value of AI solutions investigated in detecting COVID-19 pneumonia in chest CT. Finally, the authors suggested carefulness in using such tools to avoid false positive patients. This current study overcame what was suggested by the earlier article by enrolling patients with a positive RT-PCR for COVID-19 disease. One of the main advantages of automatic or semi-automatic involved parenchyma quantification is to help in stratifying patients when it comes to admission into the hospital, as it could result in lowering the cost of unnecessary hospitalizations to free up hospital beds for more critical cases. By having a more objective and quantitative measure of lung involvement since AI algorithms are less prone to subjectivity, physicians can better stratify patients for hospitalization. Patients with minimal lung involvement might be suitable for home monitoring with close follow-up, and those with more extensive involvement might require hospitalization for closer observation and potentially more intensive treatment. For example, Caruso D. et al. [14] suggested using quantitative chest CT integrated with clinical parameters to help in the accurate triaging of COVID-19 patients. The authors assessed the lung severity score by dividing the lungs into 20 regions, and each one of them was assigned a score of 0 (no lung involvement), a score of 1 (less than 50% of lung involvement) or a score of 2 (more than 50% of lung involvement). The quantitative scoring was performed with a dedicated semi-automatic tool, and the selection of a well-aerated lung was assessed in a range between −950 and −700 HU density. Additionally, Esposito G. et al. [16] proposed the Icolung tool (Icometrix, Leuven, Belgium) as a practical tool to flag high-risk patients and lower healthcare costs. The authors created a decision tree analytical model, in which they compared a routine pathway with the one using Icolung as screening tool. For example, a patient undergoes a chest CT scan and the Icolung report shows positive findings. Accordingly, the PCR test is used to confirm the illness positivity, and both devices are used to establish a decision either to send the patient home or to set different types of in-hospital care.

Moreover, the author analyzed the transmission of SARS-CoV-2 infection, expressed as the cost per avoided infection, and the in-hospital length of stay of COVID-19 patients, expressed as the cost per avoided hospital days, creating a framework that may allow physicians to make decisions on hospital policy and resource allocation. 

Therefore, as far as our knowledge extends, this is the first study that compares an automatic AI-driven lung segmentation tool and a semi-automatic one, with the visual quantitative assessment score made by radiologists. 

It is paramount to state how the combination of AI and human expertise in radiology offers a powerful approach to better assess lung parenchyma involvement. The benefits of this compound might result in improved accuracy, enhanced efficiency and earlier detection: by leveraging the strengths of both AI and human radiologists, the overall accuracy of lung parenchyma assessment can be significantly improved. AI can highlight areas of interest, while human expertise can ensure a nuanced and complete interpretation of the findings. To obtain this, clear communication between medical experts and AI developers is crucial to ensure that AI tools are designed to meet the specific needs of clinical practice. 

After all, this article presents a few limitations: Firstly, the retrospective nature of the study, since this article relies on existing medical records, which might not have been collected with the specific research question in mind. Secondly, the number of patients enrolled (*n* = 90) could lead to a selection bias of non-representativeness of the entire target population. Studies with a larger sample size might be needed to obtain more solid evidence in this field. Thirdly, the engagement of just one operator to perform the SBQAS with the semi-automatic software could intrinsically lead to measurement bias. Lastly, the HU threshold set for Icolung to detect lung abnormalities is different from the variable ones used by CT-COPD due to the nature of the deep learning process versus traditional programming, which depends on predefined rules. 

## 5. Conclusions

This study showed moderate and good agreement upon the VQAS and the SBQAS between the two software programs and the two radiologists and the consistency and reliability of the SBQAS scores across different raters. Therefore, AI should be used as a decision-making aid, not a replacement for the expertise of medical professionals. 

Finally, it is important to remember that using a combination of AI tools and physician expertise can lead to the most accurate diagnosis and treatment plans for patients.

## Figures and Tables

**Figure 1 diagnostics-14-00985-f001:**
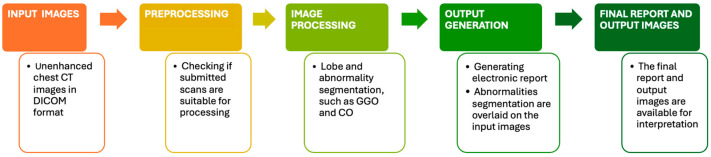
This figure explains how Icolung works.

**Figure 2 diagnostics-14-00985-f002:**
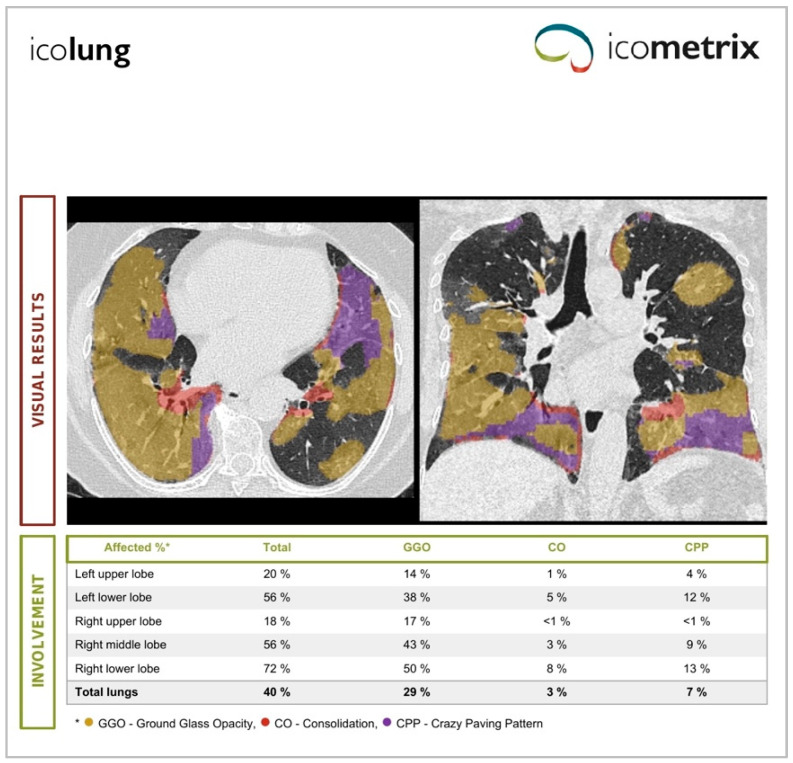
This figure shows an example of a report from Icolung (Icometrix, Leuven, Belgium).

**Figure 3 diagnostics-14-00985-f003:**
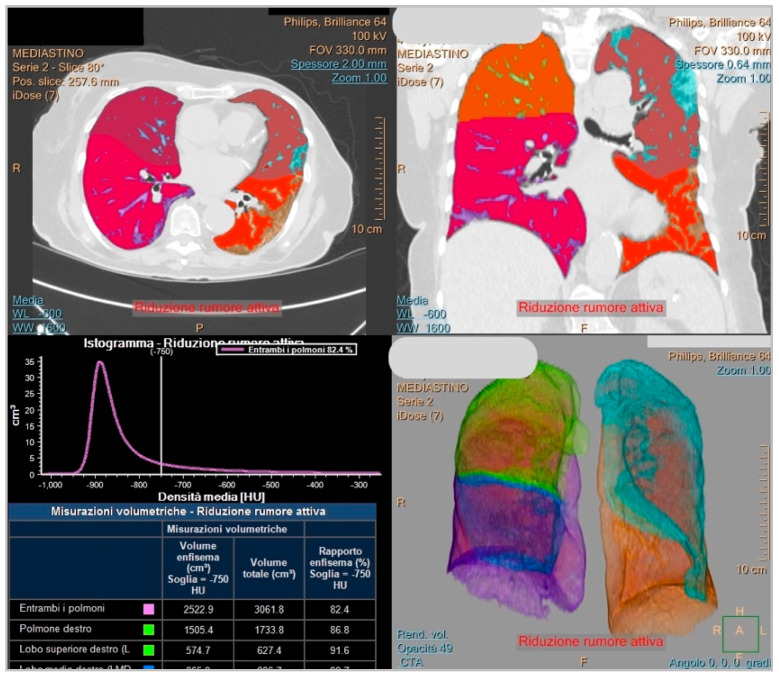
This figure displays an example of a report from COPD (Philips, Eindihoven, The Netherlands).

**Figure 4 diagnostics-14-00985-f004:**
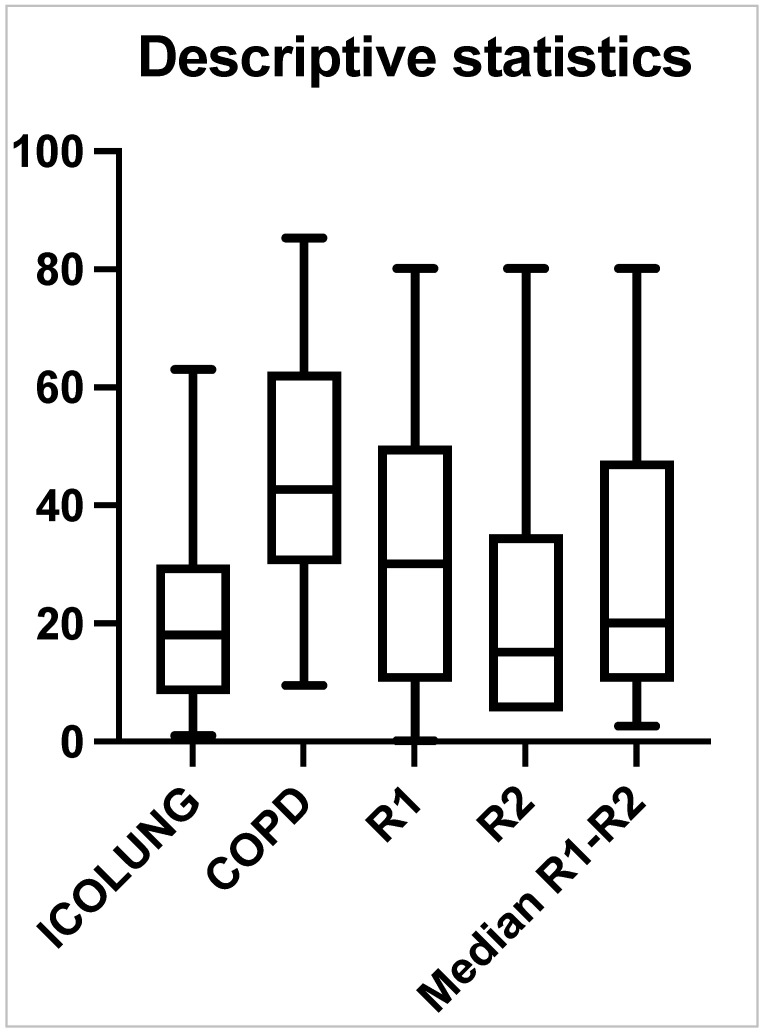
A boxplot scheme of the two software types (Icolung and COPD), the two radiologists (R1 and R2) and the median of R1–R2.

**Figure 5 diagnostics-14-00985-f005:**
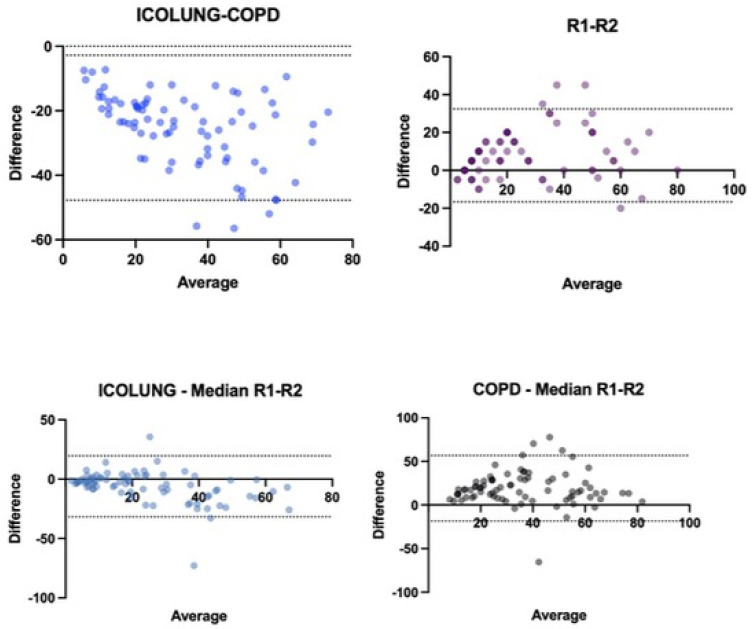
Bland–Altman graphics show the trend of the values assessed.

**Table 1 diagnostics-14-00985-t001:** This table shows the technical parameters used to acquire the chest CT scans.

Technical Parameters	Values
kV	100–120–140
effective mAs	80–350
fixed mAs	30–80
rotation time	0.4
pitch	0.8–1.2
individual detector size	0.625 mm
detector configuration	64 × 0.625 mm
thickness	2 mm
increment	2 mm

**Table 2 diagnostics-14-00985-t002:** A summary of the descriptive statistics of the two software types (Icolung and CT-COPD) and the two radiologists (R1 and R2).

	25th Percentile	Median	75th Percentile
**Icolung**	8	18	29.5
**CT-COPD**	30	42.7	62.5
**R1**	10	30	50
**R2**	5	15	35

## Data Availability

Data are available on request to the corresponding author.

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
