# Peer review of "Comparing Visual and Software-Based Quantitative Assessment Scores of Lungs’ Parenchymal Involvement Quantification in COVID-19 Patients"

_diagnostics, 2024, doi:10.3390/diagnostics14100985_

Round 1
Reviewer 1 Report
Comments and Suggestions for Authors
Nicolò et al. provide valuable insights into the comparison of visual and software-based assessment methods for lung parenchyma involvement in COVID-19 patients, emphasizing the importance of AI as a decision-making aid in radiology. As mentioned by the authors there are limitations such as the retrospective nature of the study, and the number of patients enrolled (n=90) which probably can't be amended.
The manuscript is well written although it shows some points which should be addressed to improve its quality.
Give the limitations mentioned above I suggest to:
-Regarding Visual Quantitative Assessment score: please address potential biases, such as inter-observer variability, and discuss steps taken to minimize them.
- As author acknowledged the engagement of just one operator could lead to selection biases. Please repeat with different operators blinded by the groups the SBQAS analysis with the semi-automatic software. Please provide more information on the methodology used to compare the software-based assessment scores, including statistical analyses employed.
-Have the authors tried to standardize the HU threshold across both software tools by conducting a calibration process and subsequently adopt a standardized threshold?
Comments on the Quality of English Language
Minor spell check needed.
Author Response
Dear Reviewer,
I would like to thank you for taking your time to review our manuscript. Here's our point-by point response:
- We added this paragraph at lines 162-164:
This approach was taken to minimize potential biases, such as inter-observer variability, and the efforts to minimize it was adapting a standardization of detailed scoring guidelines and a blinded scoring between the two readers.
- One of our authors (M.N) conducted a blinded SBQAS as you suggested. In response to the review's pertinent observation regarding the potential influence of operator engagement on selection biases within our study, we undertook corrective measures aimed at bolstering the methodological robustness of our Semi-Automatic Software-Based Quantitative Assessment System (SBQAS) analysis. Specifically, we replicated the SBQAS analysis utilizing the semi-automatic software, with the involvement of a distinct operator whose assessments were blinded to the group allocations. This procedural refinement sought to mitigate the influence of individual operator biases.
Subsequently, a comparative evaluation of the SBQAS assessments rendered by the two blinded operators was conducted. Notably, the Intraclass Correlation Coefficient (ICC), a widely employed metric for assessing inter-rater reliability, was computed to quantify the degree of agreement between the independently derived scores. The calculated ICC value of 0.95 obtained signifies a high level of concordance between the assessments provided by the two operators. This observation underscores the consistency and reliability of the SBQAS scores across different raters.
- We could not calibrate or standardize the HU threshold among the two software since just of them is a semi-automatic one with editing options available. You may find the reason why we used the -750 HU in reference n°8.
Reviewer 2 Report
Comments and Suggestions for Authors
1) Please define the framework of the software used. Data feeding conditions and decision making of the software at each step in a flow chart in figure would help readers understand these AI based software approach better. Right now, enough information has not been provided in this article about the software's workflow to understand its decision making ability training.
Line 108-110: Consider revising. The Tenses are varying. It should be detect. And quantification of.
Line 125: Please add a ":". The following criteria are mentioned below.
Line 137: Authors mention changing the strength between 4-7. Changing the strength could impact reconstruction parameters. Could they comment on how this can influence the correlation they have reported?
Line 138: As in it can be seen. Please delete in.
Line 155: Please revise. The sentence is confusing
Line 161: Was there a score 4?
Line 166: t returns. Is it returns.
Line 176-180: Please consider revising.
Line 229-232: Please present the exclusions in a concise manner.
Comments on the Quality of English LanguagePlease look at the comments to authors. Most of the comments are rewrite in concise manner
Author Response
Dear Reviewer,
I would like to thank you for taking your time to review our manuscript. Here's our point-by point response:
- We inserted a flowchart after line 207. An issue of proper layout in the text has occurred and we apologize for that.
- Line 108-110, we changed as follows: The first one relies on the amount of lung abnormalities visually recognized by experienced radiologists, while the second one is built upon software based on AI to automatically or semi-automatically detect lesions giving a report about the quantification of lung parenchyma involved.
- Line 125, we added what you noted.
- Line 137, we added this lines: In literature, a comparison between the application of different level of strength of iDose4 shown not significant difference in the image quality and in their interpretation [9,10].
- Line 138, we deleted what you asked.
- Line 155, we revised as follows: This scoring system analyses the affected parenchyma in each of the five lung lobes.
- Line 166, we forgot to put a “i” as the phrase results in “Moreover it returns”
- Line 176-180, we revised as follows: Deep learning algorithms are a type of machine learning inspired by the structure and function of the human brain. They achieve complex tasks by mimicking the way neurons connect and transmit information in artificial neural networks (ANNs). These are interconnected layers of processing units (artificial neurons) that loosely resemble biological neurons.
- Line 229-232, we revised as follows: This index was also used to analyze the SBQAS with the involvement of two distinct operators, whose assessments were blinded to the group allocations. This procedural refinement sought to mitigate the influence of individual operator biases. Notably, the ICC a widely employed metric for assessing inter-rater reliability, was computed to quantify the degree of agreement between the independently derived scores.
- The lines 229-232 were also noted by the other reviewer, so followed both of your comments
Round 2
Reviewer 1 Report
Comments and Suggestions for Authors
The manuscript has been improved and is now suitable for publication.
Reviewer 2 Report
Comments and Suggestions for Authors
NA
Comments on the Quality of English LanguagePlease remove the comments on the version of the paper uploaded